# CNN-Based Laue Spot Morphology Predictor for Reliable Crystallographic Descriptor Estimation

**DOI:** 10.3390/ma16093397

**Published:** 2023-04-26

**Authors:** Tom Kirstein, Lukas Petrich, Ravi Raj Purohit Purushottam Raj Purohit, Jean-Sébastien Micha, Volker Schmidt

**Affiliations:** 1Institute of Stochastics, Ulm University, 89096 Ulm, Germany; 2Université Grenoble Alpes, CEA, IRIG, MEM, 38000 Grenoble, France; 3Université Grenoble Alpes, CNRS, CEA, IRIG, SyMMES, 38000 Grenoble, France

**Keywords:** Laue microdiffraction, Laue spot morphology, Laue spot quality, convolutional neural network, polycrystalline material, runtime performance

## Abstract

Laue microdiffraction is an X-ray diffraction technique that allows for the non-destructive acquisition of spatial maps of crystallographic orientation and the strain state of (poly)crystalline specimens. To do so, diffraction patterns, consisting of thousands of Laue spots, are collected and analyzed at each location of the spatial maps. Each spot of these so-called Laue patterns has to be accurately characterized with respect to its position, size and shape for subsequent analyses including indexing and strain analysis. In the present paper, several approaches for estimating these descriptors that have been proposed in the literature, such as methods based on image moments or function fitting, are reviewed. However, with the increasing size and quantity of Laue image data measured at synchrotron sources, some datasets become unfeasible in terms of computational requirements. Moreover, for irregular Laue spots resulting, e.g., from overlaps and extended crystal defects, the exact shape and, more importantly, the position are ill-defined. To tackle these shortcomings, a procedure using convolutional neural networks is presented, allowing for a significant acceleration of the characterization of Laue spots, while simultaneously estimating the quality of a Laue spot for further analyses. When tested on unseen Laue spots, this approach led to an acceleration of 77 times using a GPU while maintaining high levels of accuracy.

## 1. Introduction

With the advent of sources, optics and detectors dedicated to X-ray characterization, X-ray diffraction techniques have become highly used tools for the quantitative studies of microstructures in many materials science and engineering applications due to their non-destructive nature and high spatial resolution. Among them, Laue microdiffraction is a spatially resolved X-ray scattering technique which is particularly sensitive to the structural arrangement of atomic lattice planes. It allows the capture of spatial maps of crystallographic orientation and the strain state of (poly)crystalline specimens [1,2]. The maps are obtained by systematically scanning the specimens with a polychromatic incident X-ray beam and the subsequent analysis of the resulting scattering patterns recorded on a planar detector. These so-called Laue patterns, as can be seen in Figure 1, consist of individual Laue spots (i.e., local maxima in the recorded image) which originate from the diffraction phenomena corresponding to geometrical reflections on the lattice planes in the crystals of the probed area in the specimen. Here, each crystal produces a characteristic pattern of spots, all of which superimpose to form the recorded Laue pattern.The precise position of the Laue spots in the experimental Laue pattern is crucial for the reliable determination of the structural crystal parameters, particularly the strain or, equivalently, the lattice parameters of the crystallographic unit cell.

More precisely, the usual workflow for analyzing a Laue pattern of a single probed position in the specimen under consideration consists of three main steps [3,4,5]: first, the local (sub-pixel) maxima in the recorded image are determined, which are assumed to be the positions of the Laue peaks (peak search routine) featuring the geometrical orientation of the corresponding reflecting lattice planes with respect to the incoming X-ray beam direction. Then, for each peak obtained in this way the corresponding crystal and reflecting planes—namely their Miller indices *hkl*—are sought (indexing routine). Finally, for each crystal, the lattice parameters of the unit cell of the crystals are refined (given a reference deviatoric strain tensor for the unit cell) by matching the expected/simulated peak positions with the observed ones from the image.

During the peak search step, instead of simply proceeding with the initial peak characterization given by the brightest pixels of a Laue spot, spot shape properties and, most importantly, the peak position have to be estimated with sub-pixel accuracy in order to keep the subsequent analyses as precise as possible. Challenges comprise polycrystalline materials with small variations of orientations and/or strains. Furthermore, the presence of a complicated spatial distribution of extended crystal defects is problematic. The reason for this is that they cause overlapping Laue patterns originating from the individual crystals and complex/non-circular Laue spots, respectively, as can be seen in Figure 2.

Defining a precise peak position for these multimodal spots that allows for a stable estimation without being affected by small changes in pixel values is hardly possible. The ideal approach would be to split them and treat their subpeaks individually. However, this would involve a complex and time-consuming spot analysis that is unfeasible in practice: currently, several thousands of highly resolved images ($4\times 10^{6}$ pixel) per dataset corresponding to a raster scan sample map have to be processed. Each of these images in turn contains more than 1000 spots corresponding to the superposition of Laue patterns, each coming from individual grains. In the near future, the demand will increase to several tens of thousands of images with $36\times 10^{6}$ pixel per image. For this reason, the analysis workflow has to be as fast as possible to keep up with the flood of data measured at synchrotron sources. Recent progress has been made with machine learning approaches to unlock and accelerate the limiting indexing step of Laue patterns [6,7], while the previous step of peak search and segmentation has to keep up in order to not be the next performance bottleneck.

Regarding the peak search step, different approaches can be implemented to extract the positions of scattering peaks from digital images. On the one hand, see, e.g., [8] employed by [9], relying on image moments [10] similar to those of bivariate probability distributions. Most importantly, the centroid (which corresponds to the intensity-weighted mean value of the spatial distribution of peak pixels) is used as an estimator for the peak position for further analyses. In order to do this, first, the region of interest (ROI) located at each Laue spot, also known as blob, has to be determined, for which the moments are then computed. The classical strategy to build the ROIs is to apply thresholds [8], but procedures based on machine learning [11] have also been proposed. While the image moments are always well defined regardless of whether the spatial distribution of pixel intensities looks like a Laue spot, it is crucial to find well-fitting ROIs that contain exactly one peak in order to obtain valid peak characteristics. Additionally, when a peak is subdivided into several overlapping components, a reliable determination of each individual subpeak is difficult to perform automatically, see, e.g., Figure 2a,b.

On the other hand, instead of using the non-parametric approach stated above, other methods are based on fitting a parametric function to the Laue spots, usually using least-squares minimization techniques. For this, mainly Gaussian functions (related to the bivariate normal probability distribution) but also Lorentzian functions (related to the Cauchy probability distribution) or combinations of the two, namely pseudo-Voigt functions, are used [12,13,14]. The parametric approach has the advantage of the descriptors of well-fitting spots being easier to interpret (provided a suitable physical or structural model). Additionally, goodness-of-fit measures can be used as estimates of the model applicability. However, a drawback of the parametric approach is that it can be rather time-consuming when the experimental Laue spot differs from the fitted function.

In the present paper, a procedure using convolutional neural networks (CNNs) is proposed to rapidly estimate the geometric descriptors of Laue spots and select high-quality peaks for a subsequent strain refinement step. While minimum human intervention is sought for the highest throughput in the Laue analysis workflow, relying on a black-box classification system for selecting peaks would make it difficult or even impossible to adapt for difficult specimens/datasets. For this reason, the neural network takes a cutout of the recorded image and returns the precise peak position and key descriptors that are essential for the quality of Laue spots, instead of simply providing a binary decision concerning whether a Laue spot is of good or bad quality. As such, the exact criterion for removing a Laue spot is still explainable and customizable based on these spot descriptors without sacrificing the computational speed.

## 2. Materials and Methods

In Laue diffraction, a crystal is irradiated with X-rays and the resulting diffraction pattern is captured on a detector. The diffraction pattern consists of a series of peaks, which correspond to the diffraction of the X-rays by the crystal lattice. To analyze the image data collected in this way, the following steps are typically performed (for example, in LaueTools [14]):1.Pre-processing: The recorded Laue diffraction image is pre-processed to remove noise and improve the contrast. This may involve techniques such as smoothing, filtering or contrast enhancement.2.Peak detection: A peak detection algorithm is used to locate the approximate positions of the peaks in the diffraction pattern. This may involve thresholding the image to identify regions of elevated intensity corresponding to the Laue spots, and then using the pixel with the highest intensity as the initial location of the potential peak for each spot.3.Peak fitting: Once the initial peak positions have been identified, parametric functions (such as a Gaussian or Lorentzian function, which are related to the bivariate normal and Cauchy probability distribution, respectively) are fitted for a precise characterization of the peaks with sub-pixel accuracy.4.Peak indexing: The positions of the peaks in the diffraction pattern are used to determine the crystal structure. This is performed by comparing the observed peak positions to the expected positions of a known crystal structure, or by using a peak-matching algorithm to determine the most likely crystal structure.5.Data analysis: The precise characterization of the peaks can be further utilized to determine the crystal structure of each individual crystal, namely the crystallographic unit cell lattice parameters or, equivalently, the strain tensor components. Compiling the results over a dataset of images collected during a sample raster scan allows the imaging of the location of crystals and crystalline defects.

Overall, the process of peak search in Laue diffraction image analysis involves several steps that are designed to identify and analyze the diffraction peaks in the diffraction pattern in order to obtain the structural parameters of experimental crystals. The present paper is concerned with the first step (peak fitting), whose outcome—the accurate characterization of peaks—strongly affects the subsequent steps. The following sections describe the employed methods and materials in detail.

### 2.1. Description of Experimental Datasets

The Laue diffraction patterns used in the present study were collected during a series of experiments conducted at the BM32 beamline at the European Synchrotron Radiation Facility (ESRF). The data comprise five datasets from a variety of materials, namely defect-free single crystals of Ge, Si, ZnCuOCl and Al_2_O_3_, as well as polycrystalline Laue patterns from materials (low-to-high absorption) with strains ranging from 0.001% to 0.2%. This includes a dataset from a thick single crystal of Al_2_O_3_, where the elongation of Laue spots is a result of depth effects. These scans were chosen to cover a range of strain levels, as well as to represent different types of crystalline structures.

The Laue diffraction images were collected using top-reflection geometry, where the 2D detector was mounted at the top of the sample and perpendicular to the incoming X-ray beam (collected scattering angles ranging from 2*θ* = 50°–130° and the sample surface was tilted by 40°. The sample-detector distance for all datasets was between 78.5 mm and 79.5 mm. The X-ray energy used in the experiments was in the range of 5–23 keV with a beam size of approximately 500 × 500 nm.

The images were recorded using a sCMOS detector with a resolution of 2016 × 2018 pixel, a pixel size of 73.4 μm and a bit depth of 16 bits per pixel. Further details regarding the experimental setup at the BM32 beamline—including the synchrotron source, the optics and beamline component—are given in [15].

### 2.2. Geometric Characterization of Laue Spots

Before the individual Laue spots can be characterized, they first have to be detected and located in the Laue image. For this reason, an initial approximation of the peak positions is obtained by the peak search algorithm implemented in the LaueTools software package, as can be seen in [14]. Specifically, connected components in the threshold Laue image, corresponding to the Laue spots, are determined and the initial peak position of each spot is then given by the position of the maximum pixel intensity or the center of mass of these regions. However, not all spots obtained in this way are useful with respect to subsequent Laue image analysis. In particular, it is important to avoid irregular or asymmetrical spots whose description by a simple single crystal model is not relevant. Including such spots in a strain refinement based on a single crystal model provides a poor average estimation of strain levels in the case of crystal defects or the assemblies of crystals. To rapidly detect these hereafter called low-quality Laue spots, a new algorithm was developed to model the shape properties of 2D Laue spots.

For that purpose, a Laue image is written as a map I:W→R, where $W\subset Z^{2}$ is a rectangular pixel array and I(v) is the 16-bit integer value of the pixel located at $v=(v_{1},v_{2})\in W$. The array of the pixel intensity values of an individual Laue spot is usually described by a 2D Gaussian function [16]. More precisely, if we let $s=(s_{1},s_{2})\in W$ be an initial guess of a Laue spot peak position, then normalization constants α,β>0 should exist such that, for small vectors $a=\left(a_{1},a_{2}\right)\in Z^{2}$, it holds that $I\left(s_{1}+a_{1},s_{2}+a_{2}\right)\approx \alpha \;\cdot \;g\left(a_{1},a_{2}\right)+\beta$. Here, $g:R^{2}\to R$ is a normalized (bivariate) Gaussian function (also known as the probability density of the bivariate normal distribution with vanishing covariances) given by
$$g\left(x_{1},x_{2}\right)=\frac{1}{2\pi \sigma_{1}\sigma_{2}}\exp\left[{\left(\frac{x_{1}-\mu_{1}}{\sigma_{1}}\right)}^{2}+{\left(\frac{x_{2}-\mu_{2}}{\sigma_{2}}\right)}^{2}\right]\;for\;all\left(x_{1},x_{2}\right)\in R^{2},$$
where $\mu_{1},\mu_{2}\in R$ and $\sigma_{1},\sigma_{2}>0$ are some location and scale parameters, respectively. Thus, to analytically describe the Laue spot at the initially predicted peak position $s=(s_{1},s_{2})$, the restriction I(A(s)) of the Laue image I:W→R to the 32 × 32 pixel cutout
$$A\left(s\right)=\left\{s_{1}-16,\cdots ,s_{1}+15\right\}\times \left\{s_{2}-16,\cdots ,s_{2}+15\right\}\subset W$$
around $s=(s_{1},s_{2})$ is considered, where I(A(s)) denotes the element-wise application of *I* on A(s), in the sense that the output is a matrix $I\left(A\left(s\right)\right)\in R^{32\times 32}$. The normalization constants α,β>0 as well as the location and scale parameters $\mu_{1},\mu_{2}\in R$ and $\sigma_{1},\sigma_{2}>0$ of the Gaussian function *g* are then fitted to the restricted image I(A(s)) using a gradient descent algorithm as implemented in the LaueTools software package [14].

In the following, we show how 2D Gaussian functions, fitted to image data, can be utilized to effectively characterize Laue spots with respect to their size, shape and position, in order to judge their usefulness for the subsequent strain analysis.

#### 2.2.1. Similarity to a Gaussian Function

As noted in [5], not all Laue spots are well described by Gaussian functions. Therefore, before analyzing the fitted parameters of a Gaussian function, we first have to verify that it fits well to the image data of the considered Laue spot candidate. For that purpose, we investigate the goodness of fit on the 32 × 32 pixel cutout A(s)⊂W introduced above. More specifically, we compute a descriptor of similarity of I(A(s)) and αg(A(s)−s)+β, given by
$${\mathrm{Sim}}^{0}\left(s\right)=\frac{\sum_{a_{1},a_{2}\in A\left(s\right)}I\left(a_{1},a_{2}\right)\;\cdot \;\left(\alpha g\left(a_{1}-s_{1},a_{2}-s_{2}\right)+\beta \right)}{\sqrt{\sum_{\left(a_{1},a_{2}\right)\in A\left(s\right)}I^{2}\left(a_{1},a_{2}\right)\;\cdot \;\sum_{\left(a_{1},a_{2}\right)\in A\left(s\right)}{\left(\alpha g\left(a_{1}-s_{1},a_{2}-s_{2}\right)+\beta \right)}^{2}}}\;.$$

However, when directly using this similarity descriptor, Laue spot candidates featuring non-Gaussian properties, such as asymmetry (see Figure 2), will still have a high ${\mathrm{Sim}}^{0}$-value. On the other hand, Laue spot candidates which appear to be well described by a Gaussian function will usually have a ${\mathrm{Sim}}^{0}$-value of above 0.95. In order to make the entire interval [0,1] descriptive with respect to the goodness of fit, we instead use the rescaled value Sim(s) to evaluate the similarity of a Laue spot at s∈W to a Gaussian function, where
$$\mathrm{Sim}\left(s\right)=\left\{\begin{array}{cc} 0, & \mathrm{if}\;{\mathrm{Sim}}^{0}\left(s\right)\leq 0.85,\\ \frac{{\mathrm{Sim}}^{0}\left(s\right)-0.85}{0.15}, & \;\mathrm{else}. \end{array}\right.$$

In cases where the Sim-value is low, the description of Laue spots using the fitted Gaussian functions is not accurate. Thus, such Laue spots should not be used for further analysis.

#### 2.2.2. Precise Peak Position

For the analysis of Laue patterns, the accurate estimation of peak positions is of great importance. For a peak located at the pixel $s=(s_{1},s_{2})$, the peak of the fitted Gaussian function is given by $\mathrm{Pos}\left(s\right)=\left(s_{1}+\mu_{1},s_{2}+\mu_{2}\right)$. This precise peak position can achieve sub-pixel accuracy to reliably locate those Laue spots which have a high Sim-value. Similarly to how the fitted location parameters $\mu_{1},\mu_{2}$ of the Gaussian function are used in order to estimate the sub-pixel peak position of Laue spots, the scale parameters $\sigma_{1},\sigma_{2}$ can be used to describe their shape and size.

#### 2.2.3. Shape and Size Descriptors

As mentioned above, it is of great importance to only consider high-quality Laue spots for strain determination in order to achieve the highest reliability. This quality does not only depend on the Gaussian similarity of Laue spots, but also on their shape and size. Hence, in general, large (even symmetrical) elongated spots are discarded. For that reason, we consider the aspect ratio and the size of the spots as featuring descriptors, which are given by
$$\mathrm{Asp}\left(s\right)=\frac{\min(\sigma_{1},\sigma_{2})}{\max(\sigma_{1},\sigma_{2})}\;and\;\mathrm{Area}\left(s\right)=\pi \sigma_{1}\sigma_{2}\;,$$
respectively. This multivariate description approach can be utilized to reject or accept spots for strain analysis. However, performing the 2D Gaussian fitting of Laue spots, as stated above, for a large number of Laue images and spots can be rather time-consuming. Thus, we propose an alternative method for the estimation of Laue spot descriptors, which is based on CNNs.

### 2.3. CNN-Based Prediction of Geometric Spot Descriptors

In order to efficiently determine the peak position Pos(s), size Area(s) and aspect ratio Asp(s) of a Laue spot with the preliminary peak position at s∈W, we will directly estimate these descriptors from the image data I(A(s)) corresponding to the Laue spot at *s*, instead of using the iterative gradient descent-based approach considered in Section 2.2.

Since the input I(A(s)) is an image, conventional methods such as linear regression, random forests or dense neural networks [17] are not suitable. In particular, these approaches do not maintain the spatial correlation of input arguments (i.e., the values I(v) for v∈A(s)), which is of great importance for analyzing image data. For that reason, CNNs, which leverage this spatial structure using convolutions [18], have been popularized in the literature. In the following, we present details regarding the specific network architecture used in the present paper. For more information on CNNs in general, we refer to [18,19].

#### 2.3.1. Adjusted Descriptors

To characterize Laue spots by means of CNNs, some of the descriptors introduced in Section 2.2, namely the size Area(s) and the peak position Pos(s), are not suitable. For this reason, these spot descriptors are adjusted. Recall that the (precise) peak position of the Laue spot at s∈W is described by $\mathrm{Pos}\left(s\right)=\left({\mathrm{Pos}}_{1}\left(s\right),{\mathrm{Pos}}_{2}\left(s\right)\right)\in R^{2}$, while the values of Sim(s) and Asp(s) belong to the interval [0,1]. Furthermore, note that a spot has low quality if the precise peak position Pos(s) deviates coordinate-wise by more than 1 from the initially estimated peak position *s*, which occurs if $|{\mathrm{Pos}}_{1}\left(s\right)-s_{1}|>1$ or $|{\mathrm{Pos}}_{2}\left(s\right)-s_{2}|>1$. This is due to the fact that the precise peak position Pos(s) is just a refinement in the sub-pixel scale of the otherwise correct initial peak position *s*. Therefore, the values of ${\mathrm{Pos}}_{1}\left(s\right)$ and ${\mathrm{Pos}}_{2}\left(s\right)$ are only of interest when $\left({\mathrm{Pos}}_{1}\left(s\right)-s_{1},{\mathrm{Pos}}_{2}\left(s\right)-s_{2}\right)\in {\left[-1,1\right]}^{2}$. If the precise peak position Pos(s) deviates coordinate-wise by more than 1 from the initially estimated peak position *s*, errors in the prediction of other descriptors could be potentially large, which would overemphasize the effect of those low-quality spots during training. Thus, instead of estimating ${\mathrm{Pos}}_{i}\left(s\right)$ for i∈1,2, we consider the adjusted position ${\mathrm{Pos}}^{★}\left(s\right)=\left({\mathrm{Pos}}_{1}^{★}\left(s\right),{\mathrm{Pos}}_{2}^{★}\left(s\right)\right)$ given by
$${\mathrm{Pos}}_{i}^{★}\left(s\right)=\left\{\begin{array}{cc} 0, & \mathrm{if}\;{\mathrm{Pos}}_{i}\left(s\right)-s_{i}\leq -1.1,\\ 1, & \mathrm{if}\;{\mathrm{Pos}}_{i}\left(s\right)-s_{i}\geq 1.1,\\ \frac{{\mathrm{Pos}}_{i}\left(s\right)-s_{i}+1.1}{2.2}, & \;\mathrm{else}, \end{array}\right.$$
for i=1,2. Note that the precise peak position Pos(s) can be reconstructed from ${\mathrm{Pos}}^{★}\left(s\right)=\left({\mathrm{Pos}}_{1}^{★}\left(s\right),{\mathrm{Pos}}_{2}^{★}\left(s\right)\right)$ for spots such that $\mathrm{Pos}\left(s\right)-s\in {\left[-1,1\right]}^{2}$. For the remaining (low-quality) spots, the precise peak position is of no interest as its estimation cannot be assumed to be reliable and these spots will thus be omitted in further analysis.

Similarly, the size descriptor Area(s), given by $\mathrm{Area}\left(s\right)=\pi \sigma_{1}\sigma_{2}$, takes values in the set of positive real numbers $R_{+}=\left(0,\infty \right)$. However, since large Laue spots are considered to be of low quality, instead of directly estimating the quantity Area(s), we consider the adjusted size
$${\mathrm{Area}}^{★}\left(s\right)=\left\{\begin{array}{cc} 1, & \mathrm{if}\;\mathrm{Area}(s)\geq 5\pi ,\\ \frac{\mathrm{Area}(s)}{5\pi }, & \;\mathrm{else}, \end{array}\right.$$
which takes values in the interval [0,1]. Again, for high-quality spots, this adjustment can be reversed. Thus, the quantitative characterization of a Laue spot at $s=(s_{1},s_{2})\in W$ is given by the descriptor vector $\mathrm{Desc}\left(s\right)=(\mathrm{Sim}\left(s\right),{\mathrm{Pos}}_{1}^{★}\left(s\right),{\mathrm{Pos}}_{2}^{★}\left(s\right),\mathrm{Asp}\left(s\right),{\mathrm{Area}}^{★}\left(s\right))$. In order to directly estimate this descriptor vector from image data, the CNN architecture stated in the next section is used.

#### 2.3.2. Convolutional Neural Network Architecture

The CNN architecture considered in this paper comprises a fully convolutional stage followed by a fully connected stage, as can be seen in Figure 3 for a schematic overview. In the fully convolutional stage, convolutional layers with a kernel size of 3×3 are iteratively applied with the goal of identifying important features in the images through the convolution with several trainable kernels. Additionally, batch normalization layers are inserted after each convolutional layer. Max-pooling layers with strides of size 2 are applied after the three blocks, consisting of two convolutional layers each. The pooling layers serve the purpose of downsampling the image size, and thus, increasing the effective field of view of the subsequent block of convolutional layers without increasing the number of weights. Since the input image has a resolution of 32 × 32 pixels, (i.e., an array of shape (32,32,1)), the output of the fully convolutional stage is a (4,4,128)-array, which is then used as the input of the fully connected stage.

The following fully connected dense layers have 112, 56, 22 and 5 neurons, respectively, where the last layer represents the final output of the CNN. The activation function of the convolutional layers and the inner dense layers is the ReLU function. The output layer uses a sigmoid function, which can only take values in the interval [0,1]. In order to use this neural network to the precise estimation of the descriptor vector Desc(s), the network first has to be calibrated via training, as described below.

#### 2.3.3. Training Procedure

Given the architecture described in Section 2.3.2, the neural network can be seen as a function $f_{\theta }:R^{32\times 32}\to {\left[0,1\right]}^{5}$ where, for a given input image I(A(s)), the output also depends on the weights $\theta \in R^{347493}$ of the network. To ensure that the network output $f_{\theta }\left(I\left(A\left(s\right)\right)\right)$ approximates the ground truth descriptor vector
$$\mathrm{Desc}\left(s\right)=(\mathrm{Sim}\left(s\right),{\mathrm{Pos}}_{1}^{★}\left(s\right),{\mathrm{Pos}}_{2}^{★}\left(s\right),\mathrm{Asp}\left(s\right),{\mathrm{Area}}^{★}\left(s\right))$$
reasonably well, we first need to find suitable weights θ by training the network. For that purpose, a stochastic gradient descent algorithm with mini-batches of size 32 is used. Specifically, the ADAM optimizer [19,20] with a learning rate of 0.001 is applied to minimize the training loss.

The training loss is characterized by means of the *d*-dimensional mean absolute error (MAE), where d=5 corresponds to the five descriptors $\mathrm{Sim}\left(s\right),{\mathrm{Pos}}_{1}^{★}\left(s\right)$, ${\mathrm{Pos}}_{2}^{★}\left(s\right)$, $\mathrm{Asp}\left(s\right),{\mathrm{Area}}^{★}\left(s\right)$ considered in this paper. In the general case of *d* descriptors for some d>0, the loss is given by
$$\mathrm{MAE}\left(y^{\mathrm{true}},y^{\mathrm{pred}}\right)=\frac{1}{nd}\sum_{i=1}^{n}{∥y_{i}^{\mathrm{true}}-y_{i}^{\mathrm{pred}}∥}_{1},$$
where $y^{\mathrm{true}}=\left(y_{1}^{\mathrm{true}},\cdots ,y_{n}^{\mathrm{true}}\right)\in R^{n\times d}$ is an ensemble of *n*-true descriptor vectors and $y^{\mathrm{pred}}=\left(y_{1}^{\mathrm{pred}},\cdots ,y_{n}^{\mathrm{pred}}\right)\in R^{n\times d}$ denotes their predictions, with the absolute value norm (also known as $L_{1}$-norm) ${\left||y|\right|}_{1}=\sum_{j=1}^{d}\left|y_{j}\right|$ for any $y=\left(y_{1},\cdots ,y_{d}\right)\in R^{d}$.

In total, 70 epochs were conducted, with 100 training steps each. To avoid overfitting, the initial model development was performed on a small subset of the first dataset, leading to the choice of various hyperparameters such as the number of epochs and the number of filters in the convolutional stage of the architecture. Additionally, in order to synthetically increase the variance in the training data, input images are shifted, rotated and reflected during training, with the corresponding adjustment of the position descriptor ${\mathrm{Pos}}^{★}$.

#### 2.3.4. Ground Truth Data

As mentioned in Section 2.1, five Laue microdiffraction scans are available for the training and evaluation of the neural network. From each of these scans, a ground truth dataset $D_{i}$, for i=1,⋯,5, is obtained by first identifying all Laue spot candidates with the initial peak search algorithm, and then, fitting a 2D Gaussian function to each spot candidate, as can be seen in Section 2.2. From this, for a spot candidate at s∈W, the true descriptor vector $\mathrm{Desc}\left(s\right)=\left(\mathrm{Sim}\left(s\right),{\mathrm{Pos}}_{1}^{★}\left(s\right),{\mathrm{Pos}}_{2}^{★}\left(s\right),\mathrm{Asp}\left(s\right),{\mathrm{Area}}^{★}\left(s\right)\right)$ is determined, which is combined with the image cutout $I\left(A(s)\right)$. Thus, in summary, the ground truth datasets $D_{1},\cdots ,D_{5}$ consist of pairs of input images and corresponding ground truth descriptor vectors for each Laue spot candidate. Since the datasets contain up to 5,000,000 Laue spot candidates, a quasi-random subsampling was conducted to limit the number of Laue spot candidates to 46,000 for each dataset.

During training, neural networks emphasize the learning of the dependencies between typical values of inputs and outputs. More specifically, these typical values appear more often during training, and thus there is more pressure (i.e., higher training loss) to correctly predict the dependency between them than for other values which occur less often. This can lead to large errors if training and test data follow different probability distributions, as shown in [21]. Unfortunately, the distribution of the similarity descriptor Sim varies significantly across the five ground truth datasets, as can be seen in Figure 4, and this is likely to be true for datasets on which the CNN is applied in the future. To offset these differences, network training is conducted on resampled datasets denoted by $D_{1}^{\mathrm{res}},\cdots ,D_{5}^{\mathrm{res}}$—in fact, as detailed in Section 3 below, only a selection of these datasets is used during training. The goal of this resampling procedure is to approximate a standard uniform distribution for Sim, meaning that there are no typical values in the training data. For that purpose, the interval [0,1] is partitioned into 12 bins and ground truth pairs (i.e., Laue spots and their descriptor vectors) are assigned to each bin with respect to their similarity descriptor Sim until they contain 2000 elements. The procedure is also stopped when no pairs remain. For the resulting resampled datasets $D_{1}^{\mathrm{res}},\cdots ,D_{5}^{\mathrm{res}}$, the histograms of the descriptor Sim are shown in the bottom row of Figure 4.

## 3. Results

The goal of this section is to evaluate the model performance on previously unseen data. Results are presented that consider all Laue spots, with a special focus on Laue spots that are well fitted by the Gaussian functions.

### 3.1. Evaluation Procedure

To evaluate the final prediction performance, the available ground truth datasets $D_{1},\cdots ,D_{5}$, as can be seen in Section 2.3.4, are split into disjoint training and test data. Note that, for the evaluation to be accurate, test and training data have to be uncorrelated. However, Laue spot candidates of a given scan tend to be highly correlated, especially when many Laue patterns are collected at several places on the same crystal during the sample raster scan. Therefore, we use the following cross-validation method to ensure that the performance measures considered in the present paper are representative: for each choice of four datasets out of the five available datasets, a CNN model is trained on the union of the resampled data (e.g., $D_{2}^{\mathrm{res}}\cup \cdots \cup D_{5}^{\mathrm{res}}$) and tested on the remaining dataset (e.g., $D_{1}$). Note that, in order to evaluate the prediction performance under real-world conditions, testing is conducted on the original dataset (e.g., $D_{1}$) that is not resampled. Thus, in summary, a total of five models—corresponding to the five so-called cross-validation folds—are built with the same architecture and hyperparameters, but trained on the different combinations of four (resampled) datasets, where the model with the number *i* uses $D_{i}$ as test data (for i=1,…,5). The learning progress of the models is illustrated in Figure 5 for both the training data and the unseen test data.

In the following, only the unadjusted spot descriptors are discussed because they are of practical concern and thus easier to interpret. Since the networks return predictions for the adjusted descriptors (i.e., ${\mathrm{Area}}^{★}\left(s\right)$ and ${\mathrm{Pos}}^{★}\left(s\right)$), we inverted the formulas given in Section 2.3.1 in order to obtain predictions for the unadjusted descriptors. Recall that the adjustments of the descriptors used for training makes them robust against outliers. However, this is no longer true for their unadjusted counterparts. For example, some badly fitted Gaussian functions lead to extremely high values of the spot size descriptor Area(s) that are much larger than the cutout A(s). Since these outliers lead to skewed error scores, they are truncated. More specifically, for a Laue spot at s∈W, the value of Area(s) is set to min{Area(s),25π}, i.e., the area of a disk with a diameter of 10 pixel. Similarly, the components of the position vector Pos(s) are set to $\max\{\min\left\{{\mathrm{Pos}}_{i}\left(s\right)-s_{i},15\right\},-16\}$ for i=1,2 to ensure that Pos(s) is inside the cutout A(s).

In order to quantify the prediction errors, the (one-dimensional) mean absolute error, as can be seen in Section 2.3.3, is considered for three different spot descriptors. Namely, ${\mathrm{MAE}}_{\mathrm{Sim}}$ for the similarity to a Gaussian function, as well as ${\mathrm{MAE}}_{\mathrm{Area}}$ and ${\mathrm{MAE}}_{\mathrm{Asp}}$ for the size and aspect ratio of the spots, respectively. Furthermore, since the peak position Pos(s) is a two-dimensional spot descriptor, the averaged Euclidean norm $\bar{\mathrm{PosErr}}$ of the position error vector is employed, which the vector obtained by subtracting the true position from the predicted position. Last but not least, the *d*-dimensional coefficient of determination $R^{2}$ is considered, where
$$R^{2}\left(y^{\mathrm{true}},y^{\mathrm{pred}}\right)=1-\frac{\sum_{i=1}^{n}{∥y_{i}^{\mathrm{true}}-y_{i}^{\mathrm{pred}}∥}_{2}^{2}}{\sum_{i=1}^{n}{∥y_{i}^{\mathrm{true}}-\bar{y^{\mathrm{true}}}∥}_{2}^{2}}$$
for an ensemble of *n* true values of a (*d*-dimensional) descriptor $y^{\mathrm{true}}=\left(y_{1}^{\mathrm{true}},\cdots ,y_{n}^{\mathrm{true}}\right)\in R^{n\times d}$ and their corresponding predictions $y^{\mathrm{pred}}=\left(y_{1}^{\mathrm{pred}},\cdots ,y_{n}^{\mathrm{pred}}\right)\in R^{n\times d}$, with the mean value $\bar{y^{\mathrm{true}}}=\frac{1}{n}\sum_{i=1}^{n}y_{i}^{\mathrm{true}}$ and the Euclidean norm ${∥y∥}_{2}=\sqrt{\sum_{j=1}^{d}y_{j}^{2}}$ for any $y=\left(y_{1},\cdots ,y_{d}\right)\in R^{d}$.

Note that the coefficient of determination $R^{2}$ relates the variation of the residuals to the variation with respect to the (single) estimator $\bar{y^{\mathrm{true}}}$. In the scalar case, i.e., for d=1, the latter is proportional to the variance in the data $y^{\mathrm{true}}$. The best possible value of $R^{2}$ is 1, and 0 is obtained if the predictive power (in the sum-of-squares sense) is equal to that of $\bar{y^{\mathrm{true}}}$, but it can become arbitrarily low (taking even negative values)—see [22] for more information regarding the scalar case. In the following, the coefficient of determination is denoted by $R_{\mathrm{Sim}}^{2}$ for the similarity to a Gaussian function, by $R_{\mathrm{Pos}}^{2}$ for the spot position, by $R_{\mathrm{Area}}^{2}$ for the spot area, and by $R_{\mathrm{Asp}}^{2}$ for the aspect ratio of the spots.

### 3.2. Numerical Results for All Laue Spots

In Table 1, the results are presented which were obtained for various error scores on the test data of each cross-validation fold as well as for aggregated error scores, where the aggregated error scores were obtained by aggregating the predictions of all five CNN models on the respective test data. It is important to emphasize that the ground truth for all predicted descriptors is directly derived from the Gaussian functions fitted to the Laue spots. In other words, for spots that can only be inadequately described by a Gaussian function, the descriptors take almost arbitrary values, which are difficult—if not impossible—to estimate directly from the image data, as can be seen in Figure 2a. This can be seen by analyzing the dependence of the similarity descriptor Sim on the prediction errors.

In Figure 6, four top views of 2D histograms are shown, corresponding to different pairs of prediction errors, where the colors indicate the heights of the histogram values at the corresponding positions. These histograms were computed using each Laue spot in the five test datasets, except those identified as outliers. Note that the prominent line in the bottom-left plot of Figure 6, corresponding to the absolute error of predicted similarity, is caused by spots whose predicted similarity is close to 0, but whose true similarity is non-zero.

### 3.3. Numerical Results for Good-Quality Spots

It is clear that the majority of large errors in Figure 6 occur for spots whose similarity descriptor Sim is (almost) equal to zero, as can also be seen in Figure 2. For this reason, a case study was conducted where a good quality spot was defined as a spot whose similarity descriptor Sim exceeds the threshold of 0.5. Furthermore, large deviations between the initial guess of the peak position s∈W and the position Pos(s) deduced from the fitted Gaussian function are also a sign of bad spots. For this reason, spots are only retained if it holds that ${∥\mathrm{Pos}\left(s\right)-s∥}_{2}\leq 0.9$. For the sake of simplicity, the criterion considered in this paper for good quality spots only depends on the similarity descriptor and the peak position, but more complex criteria based on the spot size Area(s) and/or the aspect ratio Asp(s) could also be used, depending on the nature of the datasets under consideration. Note that, in practice, the true value of the similarity descriptor Sim(s) is unknown and only the predicted value is available. Thus, a few good quality spots are misclassified as bad quality spots and vice versa, as can be seen in Table 2. By restricting the error scores presented in Table 1 to the predicted good spots, much better results are obtained, as shown in Table 3.

## 4. Discussion

Generally speaking, the following is true for all spot descriptors considered in this paper, with the exception of similarity: to accurately describe a Laue spot as seen in the image data, the underlying Gaussian function has to fit the image data reasonably well. Otherwise, descriptors assume almost arbitrary values. It is thus unsurprising that, by removing bad quality spots from Table 1, the errors go down as shown in Table 3.

However, the opposite is true for similarity, which is already well predicted on the original data. Here, the coefficient of determination $R_{\mathrm{Sim}}^{2}$ decreases if only good quality spots are retained. The reason for this is that spots which are obviously badly fitted (i.e., the “easy cases”) are excluded, leaving only the hard cases whose precise value of similarity is more difficult to predict. However, this effect is of little importance for the practical applications of the presented approach, which can be seen by the small numbers of misclassified spots, as can be seen in Table 2. Only 1.3% of spots are bad-quality spots that are misclassified as good ones. Moreover, the 2.1% of spots which were good-quality spots misclassified as bad ones usually have even less impact in practice, as there are still enough spots for an accurate peak analysis.

Considering only good-quality spots, the peak position is predicted very well, where $\bar{\mathrm{PosErr}}=0.044$ and $R_{\mathrm{Pos}}^{2}=0.973$ for aggregated error scores, see Table 3. The threshold of 0.5 pixels that would nullify the peak position refinement is clearly surpassed even for the worst dataset. In contrast, the aspect ratio is the hardest descriptor to predict for the CNN models with respect to the coefficient of determination. While this descriptor and the area are both simple functions of the scale parameters $\sigma_{1}$ and $\sigma_{2}$ of the Gaussian function (see Section 2.2.3), the first one is given as a quotient of $\sigma_{1}$ and $\sigma_{2}$ in contrast to the second one, which is proportional to their product. The quotient is much more sensitive to slight deviations of $\sigma_{1}$ and $\sigma_{2}$, making the estimation less stable compared to the product where errors might cancel out. When considering the mean absolute errors ${\mathrm{MAE}}_{\mathrm{Area}}$ and ${\mathrm{MAE}}_{\mathrm{Asp}}$, it is important to keep their respective range in mind: for the first one, values of up to 25π≈78.5 are possible (as a result of the outlier treatment), whereas for the latter, only values up to 1 occur (as a result of the definition of Asp).

When analyzing the original distributions of similarity values for each of the five datasets considered in Figure 4, it becomes clear that they differ substantially from each other, e.g., the second dataset $D_{2}$ shows a very skewed distribution where almost all spots are badly fitted by their Gaussian function. This explains the higher error scores shown in Table 1. As mentioned above this is caused by the ill-defined descriptors for badly fitted spots. When the bad-quality spots are removed from consideration, the resulting score values are more similar to those of the other datasets. Note that in the case of dataset $D_{2}$, only relatively few (175) spots remain, but recall that only a (random) subset of all Laue spots is considered in this evaluation (see Section 2.3.4).

One idea to improve the prediction performance of the CNNs might be to specifically train models for a single descriptor such as the similarity Sim. Then, instead of having a single model that predicts all descriptors of interest, multiple models have to be employed. To investigate this further, models were trained with the same cross-validation strategy, as stated in Section 2.3, but whose output is only Sim. The error scores listed in Table 4 show that there is no major improvement compared to the previously considered all-in-one models, despite the significantly increased complexity.

By performing cross-validation on different datasets, the generalization of the prediction performance has been quantified. However, there might be further aspects that impact the performance. While the crystal structure and orientation govern the position of the diffraction peak, the shape of the Laue spot depends on the local crystal misorientation distribution within the probed volume. The datasets considered in the present paper were chosen to comprise a wide variety of spot shapes from simple ones, which are well described by Gaussian functions, to very complex ones. For this reason, we assume that the predictor will work equally well on other datasets whose local crystal misorientation distribution is within the considered spectrum, irrespective of crystal structures. Nevertheless, before applying it to datasets that are outside this broad spectrum, the prediction quality should be reevaluated. Another aspect that concerns the generalization is the distance between the sample and the detector. As mentioned above the datasets considered in the present paper were acquired with a distance between 78.5 mm and 79.5 mm. Increasing this distance leads to a homothetical transform that enlarges the Laue spots, but leaves their general shape intact. The size of the cutouts (32 × 32 pixels) that are fed to the CNN models was chosen to work well for these values of the detector distance. However, for much larger values, the Laue spots might not fully fit into the cutouts and the CNN models would thus be unable to describe them accurately. In this case, the models would need to be retrained with a larger cutout size, (e.g., 64 × 64 pixels instead of 32 × 32 pixels).

One of the main advantages of the CNN-based estimator presented in this paper is its high computational speed. To make this clear, the runtime performance was evaluated by comparing it to that of the conventional gradient descent approach implemented in LaueTools, see [14]. For this purpose, 10 repetitions of analyzing 10,000 Laue spots were conducted to determine the timings reported in Table 5.

The evaluation procedure of the runtime performance consists of two setups: As the gradient descent only uses a single processor core, the CNN-based approach was also been restricted to a single core in the first setup for a fair comparison. In this case, an acceleration of 2.089 ms/0.647 ms = 3.23 is obtained, as can be seen in Table 5. In the second setup, the predictions of the neural network are computed on a GPU. Here, a speedup of 2.089 ms/0.027 ms = 77 is achieved. While the CNN-based approach is already significantly faster for the single-threaded setup, on the GPU, it can fully leverage its inherent affinity for parallel computing, resulting in massive speedups. It is also worth mentioning that the runtime for the CNN stays the same for all inputs, whereas the gradient descent algorithm takes longer for difficult test datasets with many bad-quality spots (such as $D_{2}$). The reason for this is that, for irregular spots, the initial parameters based on pixel values do not already lead to a good fit and several iterations of optimization steps must be computed. The experiments were performed on an AMD Ryzen 9 3900x CPU and an NVIDIA RTX 2080 Super GPU.

## 5. Conclusions

In this paper, we described a CNN-based method for the characterization of Laue spots (given by a pixel intensity distribution), which is an important step when analyzing Laue patterns. With the presented method, the conventional approach based on the computationally expensive fitting of parametric functions can be replaced by fast CNNs. As such, a significant acceleration (up to 77 times when using a GPU) is achieved for the prediction of geometric spot descriptors.

Using the CNN-based method, descriptors derived from the fitted Gaussian functions can be accurately estimated for Laue spots that are well described by these functions (good-quality spots). The remaining spots have little similarity to the parametric functions, and thus, descriptors derived from these fitted parametric functions assume almost arbitrary values. For this reason, such spots are not useful for the subsequent analysis. The CNN allows to quantify this similarity and, in this way, the usefulness of a Laue spot descriptors for the subsequent analysis.

While the currently predicted spot descriptors could be estimated using traditional (albeit slower) methods, the approach proposed in the present paper can also be applied to other descriptors which are more difficult to estimate by traditional methods. For example, there is the occurrence of so-called “double peaks” which occur when two Laue spots overlap. By synthetically overlapping the grayscale images of multiple Laue spots, we could synthetically generate the realistic training data of double peaks, where descriptors for both peaks are well known. These data could be used to train a neural network in order to learn the descriptors of double peaks, which would allow us to further analyze Laue spots that are currently not well described by parametric functions.

## Figures and Tables

**Figure 1 materials-16-03397-f001:**
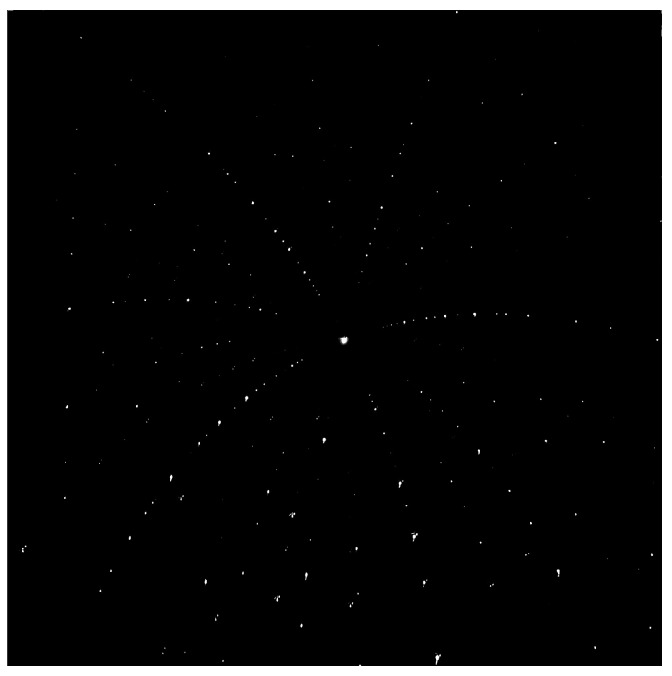
Laue pattern of a ZnCuOCl single crystal (hexagonal unit cell).

**Figure 2 materials-16-03397-f002:**
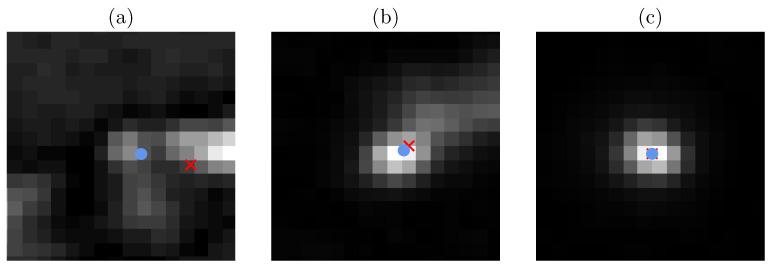
Cutouts of Laue spots of bad (**a**,**b**) and good quality (**c**) with ${\mathrm{Sim}}^{0}$-values of 0.738, 0.923 and 0.992, respectively. The corresponding (rescaled) Sim-values are 0, 0.487 and 0.952, respectively. For the definition of ${\mathrm{Sim}}^{0}$-values and Sim-values, see Section 2.2.1. Moreover, the blue dot and red cross indicate the peak position predicted by the neural network and the approach based on Gaussian functions, respectively. The cutouts are centered on the peak position predicted by the initial peak search algorithm.

**Figure 3 materials-16-03397-f003:**
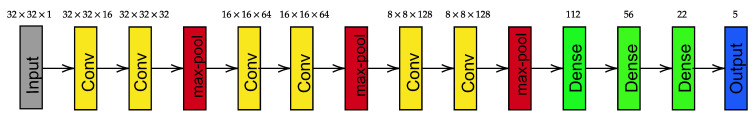
CNN architecture used for the estimation of Laue spot descriptors. The values above the convolutional layers and the dense/output layers specify the number of filters and the number of neurons, respectively. For better clarity, batch normalization layers are included in the layer Conv.

**Figure 4 materials-16-03397-f004:**
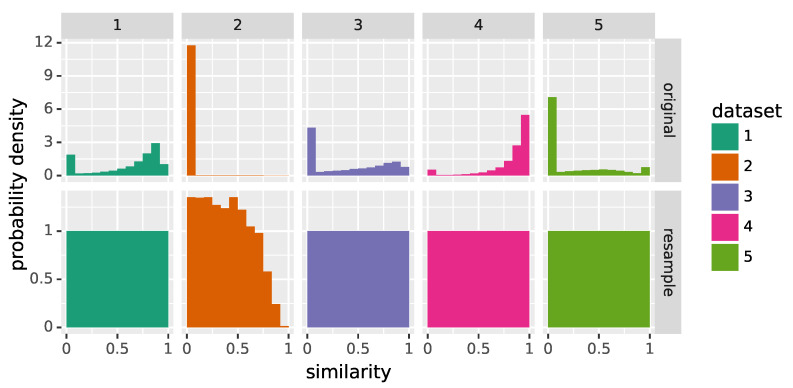
Histograms of the similarity descriptor Sim for original (**top**) and resampled datasets (**bottom**).

**Figure 5 materials-16-03397-f005:**
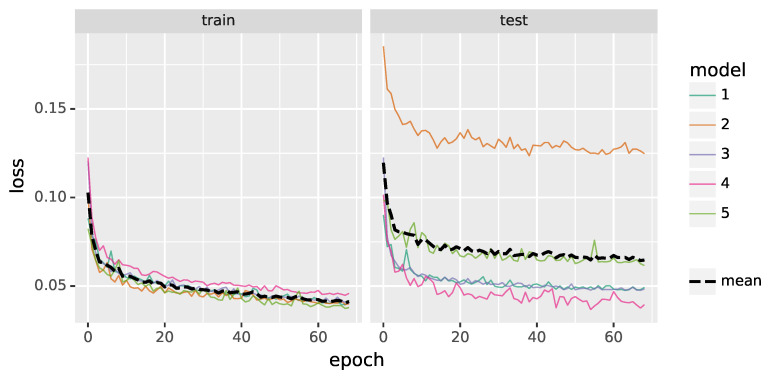
Loss functions for training data (**left**) and unseen test data (**right**) during the training of the five CNN models as well as their averaged progress (black dashed lines).

**Figure 6 materials-16-03397-f006:**
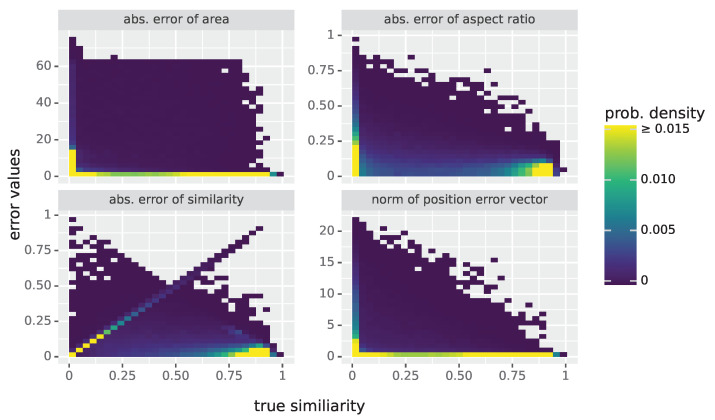
Top views of 2D histograms for different pairs of prediction errors, where the (true) similarity Sim is plotted along the *x* axis and the absolute prediction error for size (**top left**), aspect ratio (**top right**) and similarity (**bottom left**) as well as the norm of the position error vector (**bottom right**) are plotted along the *y* axis. Each underlying data point corresponds to a Laue spot in the five test datasets $D_{1},\cdots ,D_{5}$. To improve the visibility, the colormap was cut off at 0.015, and outliers were removed (instead of truncating them).

**Table 1 materials-16-03397-t001:** Results obtained for various error scores of the five CNN models on the respective test datasets, as well as for aggregated error scores.

Model	Sample Size	${\mathrm{MAE}}_{\mathrm{Sim}}$	$R_{\mathrm{Sim}}^{2}$	$\bar{\mathrm{PosErr}}$	$R_{\mathrm{Pos}}^{2}$	${\mathrm{MAE}}_{\mathrm{Area}}$	$R_{\mathrm{Area}}^{2}$	${\mathrm{MAE}}_{\mathrm{Asp}}$	$R_{\mathrm{Asp}}^{2}$
1	46,000	0.057	0.916	0.213	0.298	0.907	0.350	0.075	0.415
2	46,000	0.011	0.621	4.155	0.072	27.584	−0.264	0.184	0.047
3	46,000	0.074	0.912	0.213	0.385	0.889	0.348	0.089	0.316
4	46,000	0.051	0.889	0.051	0.819	0.249	0.318	0.055	0.260
5	46,000	0.147	0.594	2.282	0.218	25.297	−0.370	0.136	0.061
Aggregated	230,000	0.068	0.913	1.383	0.149	10.985	0.129	0.108	0.344

**Table 2 materials-16-03397-t002:** Confusion matrix for good and bad quality spots.

		True	
		Bad	Good
**Predicted**	**Bad**	0.552	0.021
	**Good**	0.013	0.415

**Table 3 materials-16-03397-t003:** Values of various error scores for *good* quality spots, computed for the CNNs on the corresponding test datasets, as well as for aggregated error scores.

Model	Sample Size	${\mathrm{MAE}}_{\mathrm{Sim}}$	$R_{\mathrm{Sim}}^{2}$	$\bar{\mathrm{PosErr}}$	$R_{\mathrm{Pos}}^{2}$	${\mathrm{MAE}}_{\mathrm{Area}}$	$R_{\mathrm{Area}}^{2}$	${\mathrm{MAE}}_{\mathrm{Asp}}$	$R_{\mathrm{Asp}}^{2}$
1	30,952	0.040	0.744	0.045	0.984	0.288	0.924	0.061	0.477
2	175	0.052	0.603	0.139	0.705	3.977	0.339	0.075	0.372
3	18,320	0.061	0.603	0.050	0.980	0.227	0.950	0.068	0.425
4	41,292	0.044	0.704	0.031	0.987	0.155	0.849	0.052	0.258
5	7633	0.100	0.530	0.091	0.838	2.396	0.500	0.075	0.315
Aggregated	98,372	0.050	0.705	0.044	0.973	0.391	0.649	0.060	0.453

**Table 4 materials-16-03397-t004:** Comparison of the error scores for CNN models that specifically predict the similarity Sim and for models combining all descriptors (Sim,Pos,Area and Asp) considered in this paper. The error scores were evaluated for the CNNs on their respective test data.

Model	${\mathrm{MAE}}_{\mathrm{Sim}}$	$R_{\mathrm{Sim}}^{2}$
specific	0.062	0.920
all in one	0.068	0.913

**Table 5 materials-16-03397-t005:** Comparison of the runtimes (in milliseconds per Laue spot) of the different approaches tested on different datasets. Note that the runtimes of the CNN-based approaches do not depend on the test datasets used.

Dataset	CNN with GPU	CNN with CPU	Gradient Descent with CPU
$D_{1}$	0.027	0.647	1.655
$D_{2}$	0.027	0.647	3.335
$D_{3}$	0.027	0.647	1.795
$D_{4}$	0.027	0.647	1.804
$D_{5}$	0.027	0.647	1.857
Aggregated	0.027	0.647	2.089

## Data Availability

Datasets of Laue patterns were collected on CRG-IF BM32 beamline at the ESRF, Grenoble, France. They are stored at ESRF and can be publicly accessed on demand and with permission according to the FAIR data policy (including potentially a 3-year embargo), as can be seen in https://www.esrf.fr/fr/home/UsersAndScience/UserGuide/esrf-data-policy.html (accessed on 3 February 2023). The latest version of the source code and the CNN model weights are available upon reasonable request to the authors.

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
