# Peer review of "CNN-Based Laue Spot Morphology Predictor for Reliable Crystallographic Descriptor Estimation"

_materials, 2023, doi:10.3390/ma16093397_

Round 1
Reviewer 1 Report
The authors reported a CNN-based method for extracting the morphology predictor of Laue spot, the manuscript is well-prepared, and the advantages of the CNN-based method are significant. Since the audience of the journal is mainly from the field of materials science, more attention should be paid to problems from the view of materials science. Before this research work is accepted, the authors need to clarify the following questions in the article:
1. The present training is conducted on datasets of Ge, Si, ZnCuOCl, and Al2O3. The crystal structure of Ge and Si is cubic, and that of ZnCuOCl and Al2O3 is trigonal. Considering the intrinsic relationship between diffraction and crystal structure, the generalization performance of the current model to the results from other crystal structures needs to be discussed.
2. Does the optic-geometry-related shape change take into consideration in mode training? The present mode focuses on the Laue spot with good quality. If I am right, that will rule out the spots far from the diffraction center, which also is valuable after geometry calibration.
Author Response
Thank you for your thorough review of our manuscript. Your comments and suggestions were very helpful in improving the quality of our work. We appreciate your time and effort, and we believe that your feedback will help us make a meaningful contribution to the field.
The authors reported a CNN-based method for extracting the morphology predictor of Laue spot, the manuscript is well-prepared, and the advantages of the CNN-based method are significant. Since the audience of the journal is mainly from the field of materials science, more attention should be paid to problems from the view of materials science. Before this research work is accepted, the authors need to clarify the following questions in the article:
1. The present training is conducted on datasets of Ge, Si, ZnCuOCl, and Al2O3. The crystal structure of Ge and Si is cubic, and that of ZnCuOCl and Al2O3 is trigonal. Considering the intrinsic relationship between diffraction and crystal structure, the generalization performance of the current model to the results from other crystal structures needs to be discussed.
The crystal structure and orientation govern the position of the diffraction peaks on the experimental Laue pattern. The shape of the peaks, that is to say the scattering intensity distribution around an average position, mostly depends on the fine local distribution of crystal misorientation within the probed volume. The crystal structure cannot be determined from a single diffraction peak area analysis, but a set of peaks (justifying the use of a 2D detector with a certain detector area). The datasets were chosen according to the variety of peak shapes irrespective of crystal structure.
We added a paragraph to the Discussion in the manuscript about the generalization of the prediction performance.
2. Does the optic-geometry-related shape change take into consideration in mode training? The present mode focuses on the Laue spot with good quality. If I am right, that will rule out the spots far from the diffraction center, which also is valuable after geometry calibration.
Increasing the distance of the detector from the sample with the same pixel size allows one to zoom in on a particular diffraction peak by increasing the angular resolution. This is a homothetical transform that affects the peak's size but not the shape. In practice, the Laue patterns are collected with a detector distance between 70 mm and 80 mm to have a high accuracy on the lattice parameters determination obtained with a large number of Laue peaks over a larger detector area. In this condition, various peak shapes and sizes are experimentally observed depending on the materials and structural defects, irrespective of peaks position with respect to the detector center.
We mention this now in the manuscript.
Reviewer 2 Report
Comments in the enclosed file

Author Response
Thank you for your feedback and recommendation regarding the publication of our manuscript. We appreciate your valuable time and effort spent on reviewing our work.
Reviewer 3 Report
The article deals with a very important subject: Laue microdiffraction is an X-ray diffraction technique. The Laue method in X- ray crystallography is more than a hundred years old and there are literally thousands if not millions of articles on the subject. It's hard to add something new, but we must. In the past scientists, let’s say, “manually” analyzed X-ray images. Today we have images containing thousands of spots for example for polycrystalline materials, as the authors rightly point out. The help of computers, neural networks and even the so-called artificial intelligence (AI) is necessary. AI would really help with this job. Of course, after careful checking and verification. Maybe in the next step.
“In the present paper, a procedure using convolutional neural networks (CNNs) is proposed to rapidly estimate geometric descriptors of Laue spots and select high-quality peaks for a subsequent strain refinement step.”
Did they succeed? In my opinion, yes. The article is long, detailed but without programming details. However, researchers interested in this subject will surely find something interesting in it.
The authors show that their method brings quite good results by analyzing (what is important) both cases of good and bad quality spots. Speeding up the process of computer image analysis is valuable, plus improving the quality of the analysis, the problem of double spots.
I have a question. Will the reader interested in your results have access to the calculation procedures? Paid or free of charge. Many scientists analyze X-ray images, also by computer. It seems that it is worth to include such information about practical use of the CNN in your article.
In my opinion, the article is interesting, contains an element of novelty and is worth publishing practically unchanged.
The only note is a loose connection between the subject of the article and the subject of the Special Issue. But as they say, everything is related to everything.
Author Response
Thank you very much for reviewing our article. We appreciate your positive evaluation of our proposed method and its potential contribution to the field.
The article deals with a very important subject: Laue microdiffraction is an X-ray diffraction technique. The Laue method in X- ray crystallography is more than a hundred years old and there are literally thousands if not millions of articles on the subject. It's hard to add something new, but we must. In the past scientists, let’s say, “manually” analyzed X-ray images. Today we have images containing thousands of spots for example for polycrystalline materials, as the authors rightly point out. The help of computers, neural networks and even the so-called artificial intelligence (AI) is necessary. AI would really help with this job. Of course, after careful checking and verification. Maybe in the next step.
“In the present paper, a procedure using convolutional neural networks (CNNs) is proposed to rapidly estimate geometric descriptors of Laue spots and select high-quality peaks for a subsequent strain refinement step.”
Did they succeed? In my opinion, yes. The article is long, detailed but without programming details. However, researchers interested in this subject will surely find something interesting in it.
The authors show that their method brings quite good results by analyzing (what is important) both cases of good and bad quality spots. Speeding up the process of computer image analysis is valuable, plus improving the quality of the analysis, the problem of double spots.
I have a question. Will the reader interested in your results have access to the calculation procedures? Paid or free of charge. Many scientists analyze X-ray images, also by computer. It seems that it is worth to include such information about practical use of the CNN in your article.
Yes, this is a very reasonable request and also very much in our own interest.
Unfortunately, it requires extensive work to prepare the source code to be deployed as a standalone program on a another computer. In contrast to uploading the implementation to a platform such as GitHub, we will therefore provide inclined readers the code as well as the trained CNN models upon request (free of charge). This way, we can give individual guidance on how to deploy the code properly. This is now also mentioned in the manuscript.
In addition, the code is planned to be deployed on the BM32 beamline at ESRF in the near future and will be available for the analysis of data from the user community.
In my opinion, the article is interesting, contains an element of novelty and is worth publishing practically unchanged.
The only note is a loose connection between the subject of the article and the subject of the Special Issue. But as they say, everything is related to everything.
Thank you for this comment. Our reasoning was that the proposed procedure advances the current state of materials characterization methods. That is why we submitted the present paper under the given subject. In any case, we are open to suggestions of a different subject from the editors.